# Assessing the Interactions of Statins with Human Adenylate Kinase Isoenzyme 1: Fluorescence and Enzyme Kinetic Studies

**DOI:** 10.3390/ijms22115541

**Published:** 2021-05-24

**Authors:** Magdalena Wujak, Anna Kozakiewicz, Anna Ciarkowska, Joanna I. Loch, Magdalena Barwiolek, Zuzanna Sokolowska, Marcin Budny, Andrzej Wojtczak

**Affiliations:** 1Faculty of Pharmacy, Nicolaus Copernicus University in Toruń, Collegium Medicum in Bydgoszcz, Jurasza 2, 85-089 Bydgoszcz, Poland; mwujak@cm.umk.pl; 2Faculty of Biological and Veterinary Sciences, Nicolaus Copernicus University in Toruń, Lwowska 1, 87-100 Toruń, Poland; anciar@umk.pl; 3Faculty of Chemistry, Nicolaus Copernicus University in Toruń, Gagarina 7, 87-100 Toruń, Poland; mbarwiolek@umk.pl (M.B.); z.a.sokolowska@gmail.com (Z.S.); awojt@umk.pl (A.W.); 4Faculty of Chemistry, Jagiellonian University, Gronostajowa 2, 30-387 Kraków, Poland; loch@chemia.uj.edu.pl; 5Synthex Technologies Sp. z o.o., Gagarina 7/134B, 87-100 Toruń, Poland; budny@synthex.com.pl

**Keywords:** adenylate kinase, atorvastatin, fluvastatin, pravastatin, rosuvastatin, simvastatin, inhibitors, fluorescence spectroscopy, pleiotropic effect

## Abstract

Statins are the most effective cholesterol-lowering drugs. They also exert many pleiotropic effects, including anti-cancer and cardio- and neuro-protective. Numerous nano-sized drug delivery systems were developed to enhance the therapeutic potential of statins. Studies on possible interactions between statins and human proteins could provide a deeper insight into the pleiotropic and adverse effects of these drugs. Adenylate kinase (AK) was found to regulate HDL endocytosis, cellular metabolism, cardiovascular function and neurodegeneration. In this work, we investigated interactions between human adenylate kinase isoenzyme 1 (hAK1) and atorvastatin (AVS), fluvastatin (FVS), pravastatin (PVS), rosuvastatin (RVS) and simvastatin (SVS) with fluorescence spectroscopy. The tested statins quenched the intrinsic fluorescence of hAK1 by creating stable hAK1-statin complexes with the binding constants of the order of 10^4^ M^−1^. The enzyme kinetic studies revealed that statins inhibited hAK1 with significantly different efficiencies, in a noncompetitive manner. Simvastatin inhibited hAK1 with the highest yield comparable to that reported for diadenosine pentaphosphate, the only known hAK1 inhibitor. The determined AK sensitivity to statins differed markedly between short and long type AKs, suggesting an essential role of the LID domain in the AK inhibition. Our studies might open new horizons for the development of new modulators of short type AKs.

## 1. Introduction

Statins are potent competitive inhibitors of 3-hydroxy-3-methylglutaryl coenzyme A (HMG-CoA) reductase. They inhibit the conversion of HMG-CoA to mevalonate, which is the rate-limiting step in the cholesterol biosynthesis [1]. For their ability to enhance the clearance of plasma low-density lipoproteins, statins have become the first-line therapy in the prevention of atherosclerotic cardiovascular disease [1,2]. These drugs possess a wide spectrum of so-called pleiotropic (cholesterol-independent) effects that may largely contribute to clinical benefits of the statin therapy and expand their use to treat other diseases [2]. In the cardiovascular system, non-lipid mechanisms of statins are associated with improved endothelial function, reduced inflammation, increased fibrinolysis and plaque stability [3]. The protective effects of statins on the vascular endothelium are related to the enhanced production and bioavailability of nitric oxide (NO) as well as restoration of the redox balance [3,4], whereas their anti-inflammatory effects include the reduction of C-reactive protein levels, leukocyte infiltration and T cell activation [5,6,7]. Moreover, statins demonstrate a promising therapeutic potential in the field of bone tissue engineering and bone regeneration due to their ability to promote the expression of bone morphogenetic proteins (BMPs) and vascular endothelial growth factor (VEGF) [8]. Statins have been shown to decrease hypertrophic scarring by reducing the expression of connective tissue growth factor (CTGF) and type I/III collagen content [9]. There is also evidence for the therapeutic potential of statins for the treatment of chronic liver disease and cirrhosis by exerting anti-fibrotic and vasoprotective effects [10]. Additionally, statins demonstrate anti-oxidant, neuroprotective, anti-viral and fungicidal effects [11,12,13,14,15]. They have also gained attention in cancer research due to their anti-tumor properties. Multiple lines of evidence indicate that certain cancers are highly dependent on the mevalonate pathway, and therefore, are vulnerable to statin treatment [16,17]. These drugs were found to inhibit the cell proliferation and induce the apoptosis in specific cancer cell types [18,19,20]. In addition, the most recent studies have shown the ability of some statins to inhibit endothelial-to-mesenchymal transition and sensitize cancer stem-like cells to chemotherapeutics [19]. It is important to emphasize that the observed in vivo and in vitro beneficial effects of the statin use are highly dependent on statin type, dose and treatment period [2,12,16,21]. Nevertheless, several observational and clinical trials demonstrated increased patient survival and reduced cancer recurrence after the statin therapy [16,20,21]. The variable pleiotropic anti-cancer effects reported for statins are to a great extent attributed to the reduced biosynthesis of isoprenoids and post-translational protein prenylation [2,16,22]. Interestingly, individual statins were found to differentially modulate gene expression profiles in cancer cells, which may explain the observed variability in anti-cancer effects of these drugs [23]. Statins are generally well-tolerated by most patients. The most common adverse effects are related to the skeletal muscles (statin myopathy). Other side effects of statins include new-onset type 2 diabetes mellitus, hepatotoxicity, renal toxicity and neurological problems [24]. The most serious problem in statin therapy is their poor water solubility and absorption after oral administration which results in a lower systemic bioavailability. To improve their therapeutic benefits, numerous nanotechnology-based drug delivery systems were developed [25,26,27,28]. The enhancement of statins’ pharmacological parameters can be achieved by using nano-sized systems and functional nanomaterials, including nanoparticles [25,27,29], nanoliposomes [9], nanocrystals [30] as well as self-nanoemulsifying systems [31,32]. Due to the wide spectrum of potential benefits of statins, research efforts should also focus on the identification of new targets and deeper understanding of the underlying mechanisms of statin pleiotropy. It could substantially improve the prognosis of patients who are administered statins, and potentially expand therapeutic options for other diseases [12].

X-ray crystallographic studies on HMG-CoA reductase in complex with statins revealed that apart from bonds formed by the HMG-like moiety, statins can form various binding interactions due to structural differences [33,34]. Type 1 statins (e.g., simvastatin, pravastatin, lovastatin) exhibit binding through the decalin-ring structure, and type 2 statins (e.g., rosuvastatin, atorvastatin, fluvastatin) demonstrate additional binding via the fluorophenyl group [33]. Studies on interactions of statins with bovine serum albumin (BSA) indicate that they have a flexible structure and their conformational changes increase the stability of the statin-BSA complex [34]. More importantly, there are some evidences for pleiotropic effects of statins via a mutual interaction with proteins, including β2 integrin leukocyte function antigen-1 (LFA-1) and P-glycoprotein (P-gp) [35,36]. Nonetheless, little attention has been paid to studying possible interactions between statins and human proteins (e.g., enzymes). This could provide a deeper insight into the pleiotropic and adverse effects of these drugs. Previous studies have shown that the P2Y_13_-mediated HDL endocytosis pathway on human hepatocytes is regulated by the cell surface adenylate kinase, indicating this enzyme as a potential target for reverse cholesterol transport regulation [37]. Statins are cholesterol-lowering drugs; however, the link between statins pleiotropy and adenylate kinase functions has been never explored. Adenylate kinases (AK, EC 2.7.4.3) are phosphotransferases regulating the nucleotide homeostasis and energetic metabolism via catalyzing the reversible phosphoryl transfer as follows: MgATP^2−^ + AMP^2−^ <=> MgADP^−^ + ADP^3−^ [38,39]. In the past decade, a growing body of evidence indicate a novel prognostic and therapeutic potential of adenylate kinases in various medical conditions [40,41]. Nonetheless, no substantial attention has been paid so far to elucidate regulatory mechanisms of the AK function and identify drugs targeting these enzymes, which could expand their clinical relevance. Notably, up to now, only one group of inhibitors, namely dinucleoside polyphosphates, was reported in the literature to inhibit some adenylate kinases [42,43,44].

For our study, we chose human adenylate kinase isoenzyme 1 (hAK1) because it is one of the most extensively studied AKs with regard to its structure, biological function and therapeutic potential. The hAK1 is expressed at the higher level in well-differentiated tissues with high energy demand, such as heart, skeletal muscle and brain [38]. AK1-catalyzed phosphotransfer is well recognized to be critical in the maintenance of cellular energetic economy and muscle performance [45,46]. In the cardiovascular system, AK1 is essential in augmenting vascular AMP signaling and adenosine production [47,48]. It was also shown to facilitate the cardiac differentiation [49] and recovery of coronary reflow following ischemia [47]. Adenylate kinase deficiency caused by mutations in the AK1 gene has been associated with severe hemolytic anemia [50] and Duchenne muscular dystrophy [51]. Apart from the protective effects driven by the aforementioned hAK1-mediated phosphotransfer, there is evidence for a pathological role of this enzyme in some diseases. The increased activity of hAK1 was found in the vitreous fluid from patients with proliferative form of diabetic retinopathy (DR), which contributes to a sustained pro-inflammatory status in diabetic eyes [52,53]. Finally, the AK1 expression is markedly increased in brains of patients with Alzheimer disease (AD), contributing to abnormal tau phosphorylation and tau-mediated neurodegeneration [54].

We investigated possible mutual interactions of statins with human AK1 using fluorescence spectroscopy approach as well as determined the impact of these drugs on the enzymatic activity of hAK1. In our study, we chose five statins representing two types of these drugs, namely pravastatin and simvastatin (type 1) and atorvastatin, fluvastatin and rosuvastatin (type 2) (Figure 1). Adenylate kinases can be classified as short type or long type, based on the differences in the LID domain present in the enzyme [55]. Conformational changes in the LID domain were found to be a rate-limiting step of the AK-catalyzed reaction [56]. In view of these findings, in addition to hAK1 as a short type AK, we also studied the effect of the selected statins on the activity of a long type AK in order to make an attempt at identifying a potential binding region of statins in AK.

## 2. Results and Discussion

### 2.1. Structure and Purity Analysis of Adenylate Kinases Used in the Study

Human AK1 has been demonstrated to play a role in several medical disorders for which a potential therapeutic use of statins has been indicated. Differences in the structure of the LID domain are the basis for the AK classification into short and long type AKs, which was related to different inhibitory efficiencies of diadenosine polyphosphates as inhibitors. Therefore, the effect of statins on the AK activity was investigated using both short and long type AKs. For the study, hAK1 as the short type and AK from *Geobacillus stearothermophilus* as the long type AK were chosen, which are well characterized adenylate kinases with regard to their kinetics and structure. Recombinant human adenylate kinase (hAK1, short type AK) and adenylate kinase from *Geobacillus stearothermophilus* (AKst, long type AK) were produced in the bacterial expression system based on the *E. coli* strain BL21-CodonPlus (DE3)-RIL and pET-3a(+) expression vector. Adenylate kinases were purified to >95% homogeneity (Appendix A). The specific activity of hAK1 and AKst in the direction of ATP synthesis (1 mM ADP as a substrate) was 87 ± 1 and 323 ± 4 U/mg of protein, respectively, whereas in the direction of ADP synthesis (AMP and ATP as substrates, 1 mM each) the activity was 463 ± 14 and 1100.6 ± 20 U/mg of protein, respectively. The specific activities of the recombinant enzymes were similar to that reported by other authors [43,57,58,59]. The CD spectrum and the calculated content of secondary structure showed that both recombinant AKs are predominately helical but with the significant content of β-sheet (Appendix A). Calculated secondary structure content was: 25.7% α-helix, 24.5% the β-sheet, 13.4% turn and 36.3% others (hAK1) and 24.8% α-helix, 11.4% β-sheet, 14.5% turn and 49.4% others (AKst). Further BeStSel analysis revealed that both enzymes belong to the alpha-beta class. These predictions are in good agreement with structural data available in PDB for hAK1 (PDB IDs: 1Z83, 2C95) and AKst (e.g., PDB IDs: 4QBH, 1ZIN), which represent alpha/beta proteins with a three-layer (aba) sandwich architecture and Rossmann fold topology.

### 2.2. Investigation of Statin–hAK1 Interactions Using Fluorescence Spectroscopy

In our study, we investigated the possible interactions of statins with human AK1 using the fluorescence spectroscopy approach. The measurements were conducted at three different temperatures: 20, 25 and 37 °C. We chose representatives of both types of these drugs, based on some clear structural differences. We used two statins of type 1, namely pravastatin (hydroxy acid form) and simvastatin (lactone form), while we chose three statins of type 2, namely atorvastatin and rosuvastatin, because they are the most different from other statins in terms of chemical structure and physicochemical properties, and fluvastatin, which is highly similar to pitavastatin [60].

Fluorescence quenching is a useful, sensitive and efficient technique for studying the conformational and/or dynamic changes of protein in the enzyme-ligand complex. Proteins contain three aromatic amino acid residues (tyrosine, tryptophan and phenylalanine) that contribute to their ultraviolet fluorescence. Absorption and emission wavelengths, as well as quantum yields, are different for these amino acid residues. Tyrosine is less fluorescent than tryptophan, but it can provide a significant signal because it exists in large numbers in proteins [61,62]. Due to the low absorptivity and fluorescence quantum yield of phenylalanine, its contribution to the intrinsic fluorescence of protein is negligible. Although human adenylate kinase isoenzyme 1 lacks tryptophan residues, it contains seven tyrosine (Y) residues (Figure 2). The intrinsic fluorescence of hAK1 from tyrosine residues can be used as an endogenous fluorescent probe to study the interactions between adenylate kinase and drugs. In our studies, the interactions between hAK1 and five statins (atorvastatin, fluvastatin, pravastatin, rosuvastatin and simvastatin) were investigated.

The hAK1 fluorescence spectra in the presence of five statins are shown in Figure 3. Our results demonstrate that with an excitation at 289 nm, hAK1 has an intense peak of fluorescent emission at 304 nm, which derives from tyrosine residues. We found that the hAK1 fluorescence intensity decreases with increasing concentration of all statins, while the maximum emission wavelength does not change. The obtained results indicate the existence of an interaction between investigated statins and hAK1, which causes the reduction in the fluorescence intensity (quenching) of the protein. Three different mechanisms of protein fluorescence quenching can usually be considered: static quenching, dynamic quenching and mixed dynamic and static quenching. All of them require molecular contact between protein and quencher. Static quenching is caused by the formation of the ground-state complex (protein-quencher complex), while dynamic quenching is caused by protein and quencher collision. The mixed quenching is caused by the collision as well as complex formation [63].

The fluorescence emission spectra of hAK1 in the presence of each statin and plot of the dependence of fluorescence intensity (F) on the statin concentration [L] at 37 °C are displayed in Figure 3. The fluorescence emission spectra and plots of F versus [L] for the hAK1 quenching process at 20 and 25 °C are presented in Appendix A, while the emission spectra of hAK1, buffer (HEPES-NaOH) and five statins are shown in Appendix A. To analyze the quenching data upon the formation of an hAK1-statin complex, we used the following equation, which assumes a 1:1 stoichiomery [64]:(1)$$\frac{\left[PL\right]}{{\left[P\right]}_{t}}=\frac{{\left[P\right]}_{t}+{\left[L\right]}_{a}+K_{d}-\sqrt{{\left({\left[P\right]}_{t}+{\left[L\right]}_{a}+K_{d}\right)}^{2}-4{\left[P\right]}_{t}{\left[L\right]}_{a}}}{2{\left[P\right]}_{t}}$$
where *K_d_* is the dissociation constant, [*P*]*_t_* is the concentration of protein, [*L*]*_a_* is the total concentration of the ligand, and [*PL*] it the concentration of protein-ligand complex.

The obtained fluorescence data were fitted to the one-site binding model with an applied nonlinear least-squares regression using OriginPro software (Figure 3). The resultant parameters obtained from the fitting procedure are shown in Table 1.

Our studies revealed that all tested statins strongly bind to hAK1, as evidenced by the K_b_ value of the order of 10^4^ M^−1^. The same strong interaction (K_b_ = 10^4^ M^−1^) was reported in the study on the interactions of statins with BSA [34]. Based on our collected results, a downward trend in the K_b_ values was observed with the increasing temperature. We observed that the binding ability of statins to hAK1 is similar at 25 and 37 °C, while the Kb values measured at 20 °C are significantly different from those determined at higher temperatures for the particular statin. It indicates that the stability of the statin–hAK1 complexes decreases with the temperature increase. We also found that the dissociation constants (K_d_) increase with the temperature increase. The K_d_ value for statins at 37 °C is in the range from approximately 15 to 30 μM. The lowest K_d_ was observed for simvastatin (15.2 ± 2.0 μM).

We performed simple docking experiments in order to predict a possible binding site of RVS to the hAK1 protein, with the use of SwissDock Server. Results indicated several different binding sites for RVS in hAK1 (Figure 4A). One of the highest-ranked regions of RVS binding to hAK1 is located in close vicinity of Y34 and Y32. However, the low binding specificity may suggest that RVS is also able to interact with other tyrosine residues, e.g., Y117 (Figure 4B). These modes of binding might explain the intrinsic fluorescence quenching of hAK1 in the presence of statins.

### 2.3. Effect of Statins on the hAK1 Activity

#### 2.3.1. Relationship between Statin Concentration and hAK1 Enzymatic Activity

In order to evaluate a possible functional implication of the discovered statin–hAK1 interactions, we studied the effect of statins on the hAK1 enzymatic activity. We found that all investigated drugs inhibited hAK1 in both directions of the enzymatic reaction in a concentration-dependent manner, although with different efficiencies (Figure 5, Table 2). The IC_50_ values for pravastatin (PVS), atorvastatin (AVS), fluvastatin (FVS) and rosuvastatin (RVS) were in the range from approximately 100 to 250 μM. Among all statins investigated, AVS showed the weakest inhibitory activity with comparable IC_50_ values higher than 200 μM in both directions of the hAK1-catalyzed reaction. FVS also exerted a comparable inhibitory effect in both directions, yet with IC_50_ values slightly higher than 100 μM. PVS demonstrated an approximately two-fold lower IC_50_ value in the presence of ADP, when compared to ATP and AMP as substrates. In turn, RVS showed a markedly higher efficiency in inhibiting hAK1 in the presence of ATP and AMP as substrates, with an IC_50_ value of approximately 90 μM. Interestingly, RVS at a concentration above 200 μM inhibited hAK1 by 90%. Finally, simvastatin (SVS) was found to efficiently inhibit hAK1 at much lower concentrations as compared to other statins. The IC_50_ value of SVS in the presence of ADP as a substrate was 5.5 ± 1.2 μM, whereas the IC_50_ value with ATP and AMP as substrates amounted to 3.1 ± 0.5 μM.

To summarize, the following order of the ability of inhibiting hAK1 by statins was observed: SVS > PVS > FVS > RVS > ATV (ADP as a substrate) and SVS > RVS > FVS > PVS > ATV (ATP and AMP as substrates). The IC_50_ values determined in the activity studies are in a good agreement with the dissociation constants (K_d_) obtained from our fluorescence experiments. Both parameters for each of the five analyzed statins are in the micromolar range.

The determined inhibitory efficiency for SVS is comparable to Ap_5_A (P^1^,P^5^-di(adenosine-5′) pentaphosphate), which is the only efficient hAK1 inhibitor discovered so far [42,44,48,65]. Ap_5_A is a bi-substrate analog with a tail-to-tail dimer structure, which, when complexed with Mg^2+^ ions, can occupy both nucleotide-binding sites of AK, and thus, act as a competitive inhibitor [66]. Depending on the source of AK (organism, cell type, body fluid), Ap_5_A was reported to completely inhibit the adenylate kinase activity at the concentration range from 10 to 200 μM [42,44,48,66,67,68]. Among all human AK isoenzymes, hAK1 was found to be abundant in blood and on the plasma membrane. For serum and endothelial cell-surface AK, the reported IC_50_ values of Ap_5_A were 6.3 μM and 6.0 μM, respectively, in the presence of 0.5 mM AMP and 0.8 mM ATP as substrates [44].

#### 2.3.2. Determination of hAK1 Inhibition Mode by Statins

Our fluorescence spectroscopy studies revealed the formation of the statin–hAK1 complex in the absence of substrates. Such observations rule out an uncompetitive inhibition mechanism where the inhibitor can bind to the enzyme-substrate complex only, and suggest a competitive or noncompetitive mechanism of hAK1 inhibition by statin [69]. In order to determine the inhibition mode, we measured the initial velocities of hAK1-catalyzed reaction at different fixed concentrations of RVS using varying ADP concentrations ranging from 0.2 to 1.0 mM. The resulting double-reciprocal Lineweaver-Burk plot yielded straight lines with different slopes intersecting on the abscissa (Figure 6A). This pattern suggests a noncompetitive mechanism of hAK1 inhibition by statins, with the apparent maximal velocity (V_max_) decreasing with increasing inhibitor concentration and the Michaelis constant (K_M_) remaining unchanged. Noncompetitive inhibitors are known to bind to the enzyme at allosteric sites (i.e., locations other than its active site). Adenylate kinase is a small monomeric enzyme consisting of CORE and LID domains and two substrate binding sites (NMPbd and NTPbd) that are specifically occupied by AMP and ATP, or alternatively by one ADP molecule each [70]. The LID domain and CORE domain with NTPbd form the ATP binding pocket, while the NMPbd and CORE domain form the AMP binding pocket [71]. A growing number of studies have provided the evidence for internal dynamics and conformational plasticity of adenylate kinases. Two subdomains (ATPlid and AMPlid) have been shown to fold and unfold in a “noncooperative manner” [72], and the largest conformational transitions occur upon substrate binding [56,73,74]. These findings suggest that the adenylate kinase activity may be regulated via allosteric interactions. Based on the results obtained from our fluorescence spectroscopy and enzyme kinetic studies, we propose that the noncompetitive inhibition of hAK1 by these drugs might involve the binding of statin in an allosteric site by either free hAK1 enzyme (E) or already complexed with the substrate (ES). This results in conformational changes of hAK1 in the active site via allosteric transitions and subsequent decrease in the apparent maximal activity of the enzyme (Figure 6B).

### 2.4. Computational Studies of AK Structure and Electrostatic Potential

Based on the length of the LID domain, adenylate kinases can be classified as short and long type [55,56]. Interestingly, previous studies showed that dinucleoside polyphosphates, in particular Ap_5_A, inhibit the short type AKs more efficiently than the long type AKs [42,43]. It is well known that the change in the LID domain position is one of the crucial steps of the reaction catalyzed by AK, and its closure enables the phosphate transfer. It was also shown that the opening of the LID domain is a step determining the catalytic reaction rate. Therefore, in view of a crucial role of the LID domain in the AK catalysis and inhibition by Ap_5_A, we also included in our studies AK from *Geobacillus stearothermophilus* (AKst), which is a well-characterized long type AK. The superposition of hAK1 and AKst shown in Figure 7 revealed that the three-dimensional structures of these enzymes are almost identical. The most apparent difference is the length and position of the LID domain, which in AKst is longer by 27 amino acids as compared with hAK1 (Figure 7C).

Next, we measured the enzymatic activity of AKst in the presence of RVS and SVS at the same drug concentration range as in the hAK1 experiments (Appendix A). Surprisingly, we have not found any inhibitory effect of these statins on the AKst activity. These data clearly show a different sensitivity of short and long type AKs to statins. Furthermore, this suggests that statins may bind to hAK1 at the region near the entrance to the active site close to the LID domain, and in this way, inhibit its activity. This region has limited accessibility in AKst due to the presence of a markedly bigger LID domain as compared to hAK1, which may consequently prevent the binding of statins to AKst.

The calculated electrostatic potentials of investigated AKs indicate that the active site pocket (where Ap_5_A is bound) is positively charged (Figure 8). The region of positive potential in the active site is present in both enzymes; however, in hAK1, it is permanently exposed to the solvent, while in AKst it is covered by the LID domain when the enzyme is in the closed state. As the statins have the negatively charged carboxyl group, it is highly probable that they are bound to hAK1 at the entrance to the active site. The lack of inhibitory effects of statins on AKst might be associated with the LID domain structure and its flexible conformational changes, which prevent a stable binding of statin at the entrance to the active site, and thereby, significantly reduce the inhibitory potential of the drug.

To conclude, the obtained results show that statins decrease the catalytic activity of the short type AK (hAK1) due to allosteric mechanisms and a potential allosteric binding site could be located within the LID domain. Therefore, the design of new modulators of short type AK might be based on the identification of amino acids of the LID domain, which are involved in allosteric communication with the statins. This would allow to propose new structural modifications of statins, resulting in the development of new efficient short type AK inhibitors.

In addition to lipid-modulating properties, statins have a large number of beneficial effects, including anti-inflammatory and cardio- and neuro-protective [3,7,11,12]. Human AK1 has a well-recognized role in maintaining the heart and skeletal muscle energy metabolism and cardiovascular function [45,46,47,48,49]. In particular, knockout mice lacking AK1 were found to demonstrate a reduced energetic efficiency of contractile performance [45,46]. Statin myopathy is related to disturbances in skeletal muscle performance, energy metabolism and exercise intolerance [24,75]. Thus, the adverse effects of statins, in particular myopathy, might be related to the hAK1 inhibition in skeletal muscle, resulting in impaired cellular energetics and muscle performance. In addition, statin–hAK1 interactions might be considered as unfavorable in the therapy of cardiovascular disorders, including ischemia/reperfusion injury, since the lack of heart AK1 was found to blunt energetic signal communication and compromise post-ischemic coronary reflow [47]. On the other hand, we have identified some medical disorders where the inhibition of hAK1 by statins might be related to some beneficial effects. hAK1 is increased in brains of patients with Alzheimer disease and is involved in β-amyloid-induced hyper-phosphorylation of tau, indicating a neuropathogenic role of this enzyme [54]. Interestingly, there is evidence for a potential therapeutic role of statins in targeting Aβ-mediated neurotoxicity [15]. Moreover, statin therapy has been associated with a lower incidence of neurodegenerative diseases, including Alzheimer’s disease [76]. Finally, the use of statins was related to a decreased prevalence of vision-threatening diabetic retinopathy [77], where a role of soluble AK1 in maintaining high pro-inflammatory ATP levels in diabetic retinopathy eyes has been recently recognized [52,53]. A possible link between the hAK1 inhibition by statins and some pleiotropic and adverse effects of these drugs merits further investigation in order to assess a clinical relevance of the discovered statin–hAK1 interactions.

## 3. Materials and Methods

### 3.1. Reagents

Atorvastatin sodium salt (>98%), pravastatin sodium salt (≥98%), fluvastatin sodium salt hydrate (≥98%) and simvastatin (≥98%) were purchased from Cayman Chemical (Ann Arbor, MI, USA). Rosuvastatin sodium salt (98%) was synthesized by Synthex Technologies Sp. z o. o. according to previously reported procedure [40]. Potassium phosphate dibasic (K_2_HPO_4_) (≥98%) and potassium phosphate monobasic (KH_2_PO_4_) (≥98%) were purchased from J.T. Baker (New Jersey, NY, USA). Disodium edetate dihydrate (EDTA) (>98.5%), 4-(2-Hydroxyethyl)piperazin-1-ylethanesulfonic acid (HEPES) (99%), ethanol (99.8%), perchloric acid (HClO_4_) (70%), magnesium chloride hexahydrate (MgCl_2_ × 6H_2_O) (99%) and potassium hydroxide (KOH) were purchased from Avantor Performance Materials (Gliwice, Poland). Adenosine 5′-monophosphate disodium salt (AMP) (≥99.9%), adenosine 5′-diphosphate sodium salt (ADP) (≥95%), adenosine 5′-triphosphate disodium salt hydrate (ATP) (99%), tetrabutylammonium hydrogensulfate (TBA) (97%), methanol (≥99.9%), isopropyl thiogalactopyranoside (IPTG), bacterial cell culture reagents and supplements for adenylate kinases production, BL21-CodonPlus (DE3)-RIL strain and pET-3a(+) expression vector were purchased from Sigma-Aldrich (Poznań, Poland).

### 3.2. Bacterial Production and Purification of Adenylate Kinases

The cDNA sequences encoding *Geobacillus stearothermophilus* AK (AKst) and human adenylate kinase isoenzyme 1 (hAK1) were synthesized *de novo* by GeneCopoeia, Inc. (Rockville, MD, USA) and GeneArt™ Gene Synthesis Service (Thermo Fisher Scientific Inc., Waltham, MA, USA), respectively. Additionally, the sequences encoding thrombin recognition site followed by 6xHis-tag were incorporated to 3′ end of hAK1 cDNA sequence. Each cDNA construct was cloned into pET-3a(+) vector between the NdeI and BamHI restriction sites. Recombinant plasmids pET-3a(+) were transformed into *E. coli* BL21-CodonPlus(DE3)-RIL cells using the heat shock method. A single transformant colony was used to inoculate 10 mL of LB medium (0.5% yeast extract, 1% tryptone, 1% NaCl, pH 7.0) containing 50 µg/mL ampicillin, 34 µg/mL chloramphenicol and 0.5% glucose. The culture was incubated overnight at 30 °C with shaking. The bacterial production of AKst was performed as previously described [59], whereas hAK1 was produced according to the method established by the authors. A total of 10 mL of the bacterial culture was transferred to 1.0 L of fresh LB medium supplemented with components as above and grown in the same conditions until an OD_600_ of 0.4–0.6 was reached. Then, the hAK1 synthesis was induced with 0.5 mM IPTG and the cells were cultured overnight at 18–20 °C with shaking. Then, the bacterial culture was cooled on ice and centrifuged at 4000× *g* for 20 min at 4 °C. The cell pellet was resuspended in 10 mM HEPES-NaOH, pH 7.5 containing 15% glycerol, 0.2 M NaCl, 10 mM DTT and 30 mM imidazole (buffer A), which was sonicated and the bacterial lysate was centrifuged at 20,000× *g* for 20 min at 4 °C. The obtained supernatant was used for the recombinant protein purification.

hAK1 and AKst enzymes were purified using ÄKTA Start Protein Purification System (GE Healthcare, Chicago, IL, USA). AKst was partially purified by the two-step heat treatment of the bacterial supernatant as previously described [59]. The obtained protein fraction was subsequently subjected to HiTrap Blue HP 5 mL column (GE Healthcare, Chicago, IL, USA) and AKst was eluted with 2 M NaCl in 25 mM HEPES-NaOH buffer, pH 7.5. The hAK1 purification was performed on HisTrap FF Crude 5 mL column (GE Healthcare, Chicago, IL, USA) equilibrated with buffer A using two step-gradient elution at 0.15 and 0.5 M imidazole in 10 mM HEPES-NaOH, pH 7.5 containing 15% glycerol, 0.2 M NaCl, 10 mM DTT (buffer B). 1 mL fractions collected at elution with 0.5 M imidazole were then loaded onto the HiTrap Desalting Column (GE, Healthcare) equilibrated with buffer B in order to remove imidazole from the protein samples. The protein content was measured using Bradford Method Protein Assay (VWR Life Science, Gdańsk, Poland). The purity of the protein was assessed by SDS-PAGE and Coomassie staining. Protein samples were incubated for 20 min at 85 °C in reducing buffer containing 4% SDS and 10% β-mercaptoethanol, centrifuged at 5000× *g* for 1 min at room temperature and subsequently run through 4% stacking gel and 12% separating gel (Bio-Rad, Mini-Protean^®^ TGX Stain-FreeTM Gels).

### 3.3. Circular Dichroism Analysis

To analyze the conformation of the recombinant hAK1 and AKst, far UV circular dichroism spectra (190–250 nm) were recorded with JASCO J-815 spectropolarimeter at room temperature using 1 mm optical path length and 3.3 μM hAK1 and 19.2 μM AKst in 50 mM phosphate buffer, pH 7.5. The secondary structure content was calculated with the use of BeStSel server [78].

### 3.4. Fluorescence Studies

The fluorescence spectra of human kinase adenylate isoenzyme 1 (hAK1) in the absence and presence of five statins (pravastatin, simvastatin, atorvastatin, fluvastatin and rosuvastatin), in 25 mM HEPES-NaOH (pH 7.5)were performed on JASCO FP-8300 spectrofluorimeter with 10 mm quartz cell (Hellma Analytics, Müllheim, Germany). Measurements were recorded in the range of 300–600 nm after excitation at λ = 289 nm at 20, 25 and 37 °C. Stock solutions of all statins were prepared in ethanol. The samples were prepared in 2 mL Eppendorf tubes and contained hAK1 at a concentration of 2 µM alone or with a statin at the following concentrations: 2, 4, 8, 16, 24, 32 and 40 µM, and 25 mM HEPES-NaOH (pH 7.5) was added to each tube, up to 2 mL. Then, the sample was transferred into the quartz cell and the spectrum was recorded. The emission spectra were measured three times. The fluorescence spectra of hAK1 in presence of statins were corrected for the inner filter effect using the following equation [63,64]:(2)$$F_{cor}=F_{obs}*\;10^{\frac{\left(A_{\lambda Ex}+A_{\lambda Em}\right)}{2}}$$
where *F_cor_* is the corrected fluorescence intensity and *F_obs_* is the observed fluorescence intensity, *A_λEx_* is the absorption value at the excitation wavelength and *A_λEm_* is the absorption value at the emission wavelength. The fluorescence data were fitted by applying a nonlinear least-squares regression using OriginPro software Version 2016 (OriginLab Corporation, Northampton, MA, USA).

### 3.5. UV–Vis Spectra Measurements

The UV–Vis absorption spectra were recorded on Eppendorf BioSpectrometer^®^ basic with 10 mm quartz cell (Hellma Analytics, Müllheim, Germany) in the range of 200–500 nm with 25 mM HEPES-NaOH (pH = 7.5) as a blank.

### 3.6. Adenylate Kinase Activity Assay and Enzyme Inhibition Studies

The reaction mixture for ATP synthesis contained 2 mM MgCl_2_, 1 mM ADP in 10 mM HEPES-NaOH, pH 7.5. The reaction mixture for ADP synthesis was composed of 2 mM MgCl_2_, 1 mM AMP and ATP each, in 10 mM HEPES-NaOH, pH 7.5. The effect of AVS, FVS, PVS, RVS and SVS on the hAK1 activity was tested in the concentration range from 0.05 to 1.6 mM. Studies on the statin inhibition mode were performed using varying concentrations of ADP as a substrate (from 0.2 to 1.0 mM) at different fixed concentrations of statin (0, 150, 300 and 450 µM). Adenylate kinase activity measurement was carried out in 50 μL of the reaction mixture for 5–10 min at 37 °C. The reaction was terminated by the addition of 50 μL of 1 M HClO_4_. After 5 min incubation on ice, samples were centrifuged at 15,000× *g* for 5 min at 4 °C. Then, 50 μL of the obtained supernatant was neutralized with 25 μL of 1 M KOH and delipidated by shaking with n-heptane (1:5 *v*/*v*). The samples were centrifuged as above, and subsequently, adenine nucleotide concentrations were analyzed by the RP-HPLC (reversed-phase high-performance liquid chromatography) method as previously described [79]. Briefly, purine nucleotides were separated in isocratic conditions on the Chromolith Performance RP-18e column, 4.6 mm × 100 mm (Merck, Darmstadt, Germany) using the Shimadzu Prominence LC (Kyoto, Japan) equipped with pump (LC-20AD), autosampler (SIL-20AC), column thermostat (CTO-10AS) and photodiode array detector (SPD-M20A). HPLC system control, data acquisitions and processing were done with LabSolutions software. The samples were eluted using 0.1 M phosphate buffer, pH 7.0 containing 25 mM TBA, 5 mM EDTA and 3% methanol. ADP and ATP produced during the AK-catalyzed reaction were detected at λ = 260 nm based on the retention times of standard nucleotide solutions. The AK enzymatic activity was determined by quantifying the product concentration using correlation coefficients calculated from ADP and ATP calibration curves. One unit (U) of AK activity is defined as the amount of the enzyme that catalyzes the production of 1.0 μmole of ADP or ATP per minute at 37 °C, under the assay conditions specified above. The specific activity of AK was calculated as the number of units per milligram of protein. The Lineweaver–Burk plot was used to present enzyme-substrate reactions with and without statin, where the y-intercept is equivalent to the inverse of *V_max_* and the x-intercept represents −1/K_M_. The calculation of the IC_50_ values was performed using GraphPad Prism 8 Software (San Diego, CA, USA).

### 3.7. Ligand Docking and Electrostatic Potential

All calculations were made utilizing coordinates of human and *Geobacillus stearothermophilus* adenylate kinases available in the Protein Data Bank. The surface charge distribution for hAK1 (PDB ID: 1Z83) and AKst in the open and closed state (PDB ID: 1ZIN and 4QBH) was calculated using PyMOL [80]. APBS plugin v2.1.0 (Adaptive Poisson-Boltzmann Solver and the PDB2PQR) was used with the default values [81,82]. Molecular docking of rosuvastatin to hAK1 was done with the use of the SwissDock server using the default server values and protein coordinates available in PDB (PDB ID: 1Z83). Coordinates of RVS were extracted from the structure 1HWL available in PDB. The results of the blind docking (with no specified binding region) were analyzed and visualized in the UCFS Chimera [83].

### 3.8. Statistical Analysis

The experiments were conducted at least two times in triplicate and qualitatively identical results were obtained. The data are expressed as means ± SEM. Grouped comparisons were carried out by two-way ANOVA. Differences with *p* < 0.05 between the groups were considered significant. All statistical analyses were performed using GraphPad Prism 8 Software (San Diego, CA, USA).

## 4. Conclusions

Our research revealed for the first time that statins can strongly bind to human adenylate kinase isoenzyme 1. Moreover, the significant novelty of the reported research is that all investigated statins exerted inhibitory effects on the hAK1 activity, yet with different efficiencies. The largest inhibitory effect was observed for simvastatin. Importantly, the calculated IC_50_ values for SVS were comparable to Ap_5_A, which is the only hAK1 inhibitor known so far. Interestingly, in contrast to the competitive inhibition mode of Ap_5_A, statins appear to be noncompetitive inhibitors of hAK1. Furthermore, our studies revealed a different sensitivity of short and long type AKs to statins, by demonstrating no inhibitory effect of these drugs on the long type AK from *G. stearothermophilus*. The performed analysis of the structures suggests that the inhibitory mechanism of statins found for short type AK might be related to the structural and conformational heterogeneity of the LID domain. We believe that the identification of statins as hAK1 inhibitors might open new perspectives for the basis for the future SAR (structure–activity relationship) analysis and design of selective modulators of short type AKs, and thereby, expand the therapeutic potential of these enzymes. The observed inhibitory effects of statins on the hAK1 activity might be considered as new possible pleiotropic or adverse effects of these drugs with a potential clinical significance in some medical settings, including neurodegenerative and inflammatory diseases as well as statin-induced myopathy. Therefore, our work might be a starting point for further studies on the medical implications of discovered statin–hAK1 interactions.

## Figures and Tables

**Figure 1 ijms-22-05541-f001:**
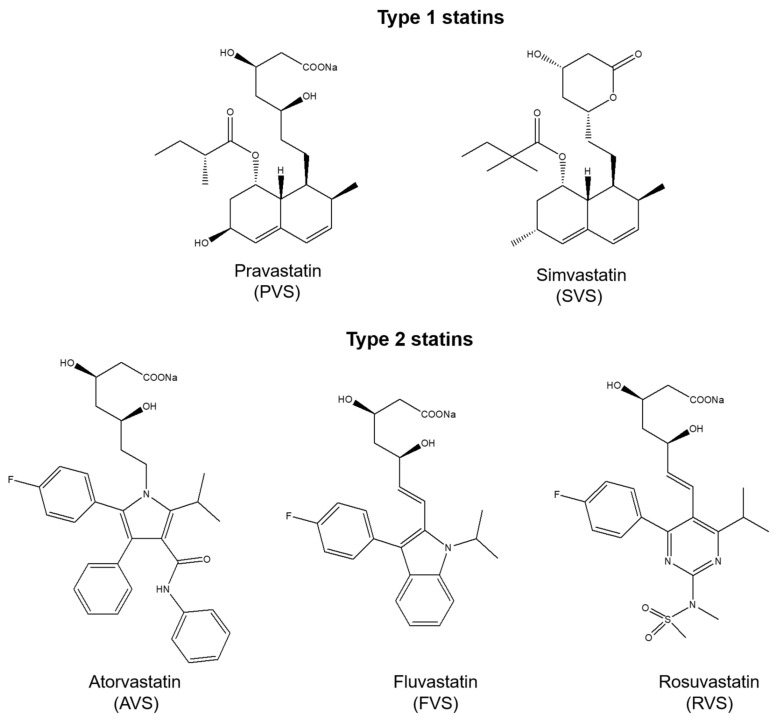
Chemical structures of statins used in the study.

**Figure 2 ijms-22-05541-f002:**
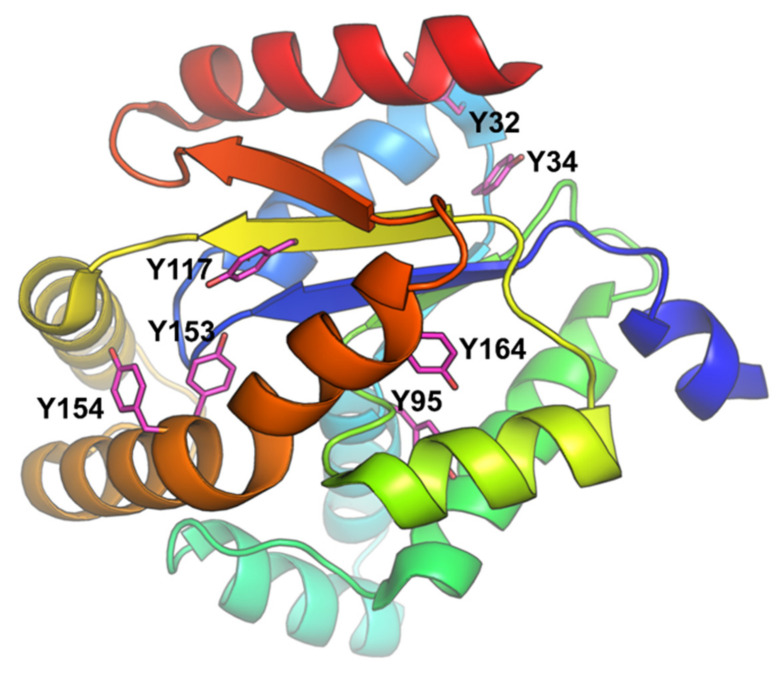
hAK1 structure (PDB ID: 1Z83) with the depicted tyrosine residues.

**Figure 3 ijms-22-05541-f003:**
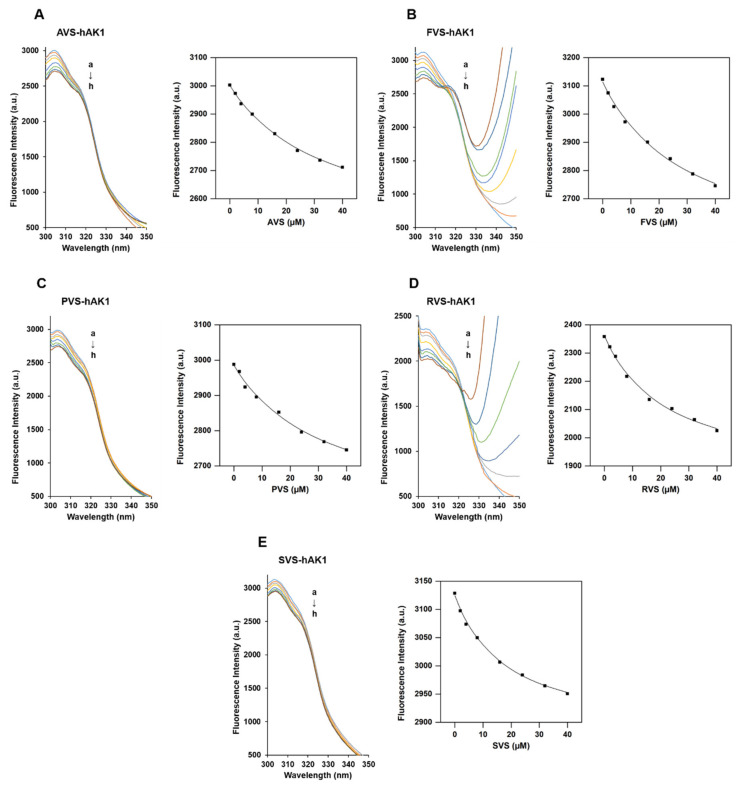
The fluorescence quenching of hAK1 in the presence of (**A**) atorvastatin, (**B**) fluvastatin, (**C**) pravastatin, (**D**) rosuvastatin and (**E**) simvastatin at 37 °C. (left) Emission spectra of hAK1 in the presence of statin, (right) dependence of fluorescence intensity on statin concentration. The concentration of hAK1 was 2 μM. The concentrations of the five statins from a to h were 0, 2, 4, 8, 16, 24, 32 and 40 μM, respectively.

**Figure 4 ijms-22-05541-f004:**
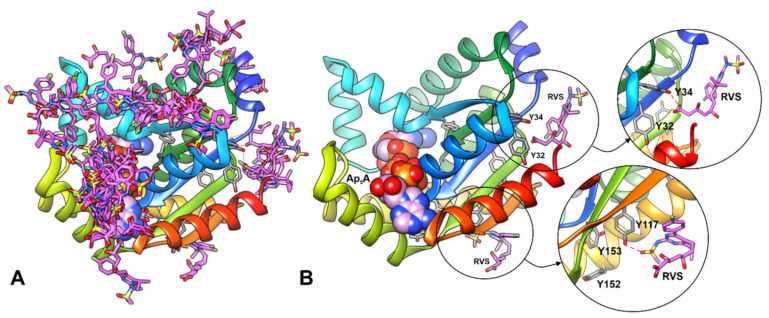
Results of blind docking of RVS to hAK1 performed with the SwissDock server. (**A**) All predicted binding sites for RVS (pink sticks) in hAK1. (**B**) The conformation of RVS at the selected possible binding regions in close vicinity of tyrosine residues. Ap_5_A is shown as spheres; possible hydrogen bonds are marked in red dashed line.

**Figure 5 ijms-22-05541-f005:**
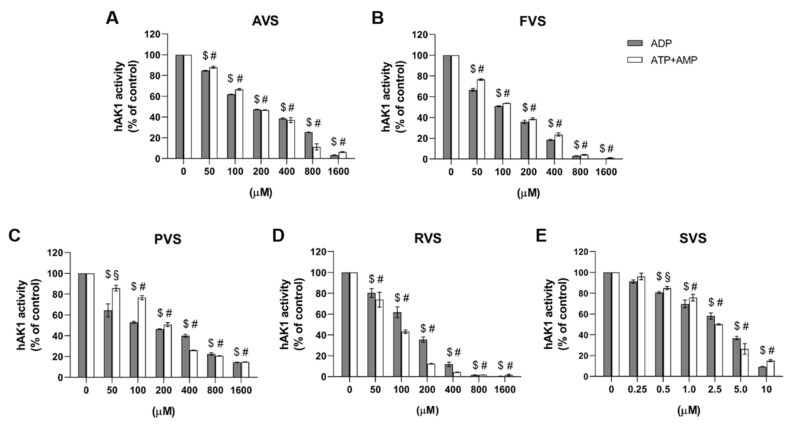
Effect of (**A**) atorvastatin, (**B**) fluvastatin, (**C**) pravastatin, (**D**) rosuvastatin and (**E**) simvastatin on the hAK1 enzymatic activity in the presence of ADP or ATP and AMP as substrates, 1 mM each. $ *p* < 0.0001 compared to the control with ADP, # *p* < 0.0001 and § *p* < 0.001 compared to the control with ATP and AMP as substrates.

**Figure 6 ijms-22-05541-f006:**
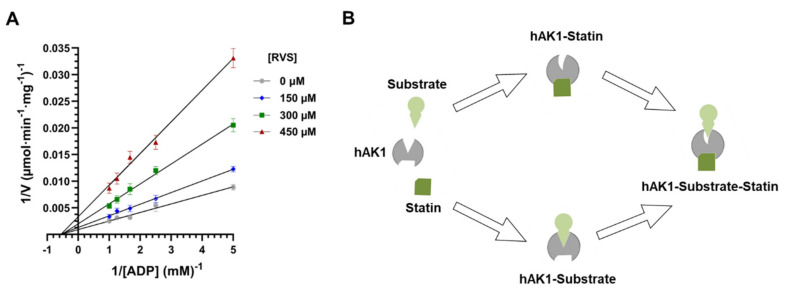
(**A**) Double reciprocal plots of hAK1 inhibition by rosuvastatin (RVS). (**B**) A graphical presentation of hAK1 noncompetitive inhibition by statins.

**Figure 7 ijms-22-05541-f007:**
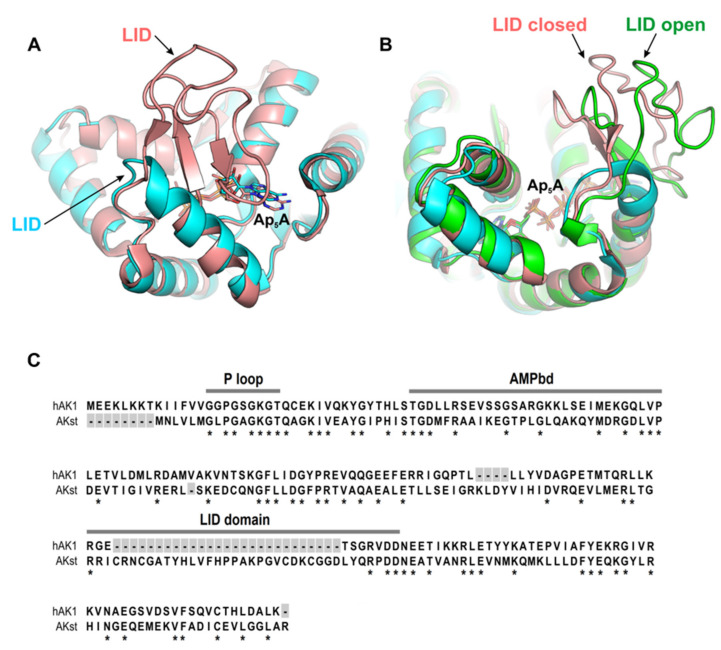
(**A**) Superposition of hAK1 (cyan) (PDB ID: 1Z83) and AKst (light pink) (PDB ID: 4QBH) showing the main structural difference between these two AKs. (**B**) The LID domain in AKst in a closed (light red, PDB ID: 4QBH) and open (green, PDB ID: 1ZIN) state. (**C**) Comparison of amino acid sequences of AKs with marked functional domains AMPbd and LID and the NTP-binding site (P loop). Asterisk indicates a conserved amino acid residue at each alignment position.

**Figure 8 ijms-22-05541-f008:**
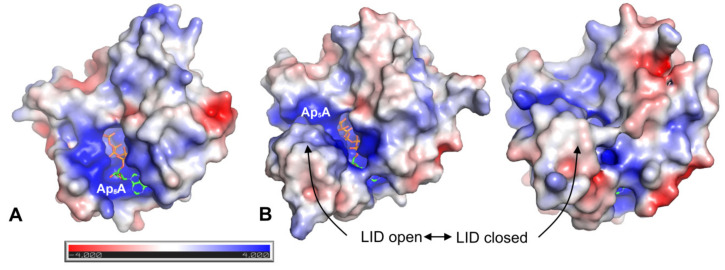
Electrostatic potential mapped on the solvent-accessible surface of (**A**) hAK1 (PDB ID: 1Z83) in the closed state and (**B**) AKst in the open (PDB ID: 1ZIN) and closed (PDB ID: 4QBH) state. Cofactor Ap_5_A is shown in sticks representation. Electrostatic potential was calculated using APBS plugin 2.1 to PyMOL.

**Table 1 ijms-22-05541-t001:** Binding parameters of hAK1 at 20, 25 and 37 °C.

Statin	T (°C)	K_b_ [L·mol^−1^] × 10^4^	K_d_ [mol·L^−1^] × 10^−6^
AVS	20	6.3 ± 0.8	16.0 ± 2.1
	25	3.7 ± 0.9	27.2 ± 6.8
	37	3.4 ± 0.4	29.5 ± 3.7
FVS	20	5.8 ± 0.6	17.4 ± 2.0
	25	4.3 ± 0.8	23.1 ± 4.0
	37	3.8 ± 0.8	26.6 ± 5.5
PVS	20	5.2 ± 1.0	19.4 ± 4.1
	25	3.8 ± 0.9	26.4 ± 6.2
	37	3.5 ± 1.0	28.6 ± 8.2
RVS	20	11.9 ± 2.0	9.2 ± 1.5
	25	5.9 ± 0.6	16.8 ± 1.8
	37	5.3 ± 0.9	18.9 ± 3.2
SVS	20	15.2 ± 2.7	6.6 ± 1.2
	25	9.2 ± 1.3	10.9 ± 1.6
	37	6.6 ± 0.9	15.2 ± 2.0

**Table 2 ijms-22-05541-t002:** IC_50_ values of statins determined from hAK1 inhibition studies at 37 °C.

Statin	IC_50_ [µM] (ADP as a Substrate)	IC_50_ [µM] (ATP + AMP as Substrates)
AVS	223 ± 34	237 ± 31
FVS	126 ± 7	156 ± 11
RVS	166 ± 20	89 ± 15
PVS	95 ± 19	211 ± 33
SVS	5.5 ± 1.2	3.1 ± 0.5

## Data Availability

The data presented in this study are available in article.

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
