# Peer review of "Assessing the Interactions of Statins with Human Adenylate Kinase Isoenzyme 1: Fluorescence and Enzyme Kinetic Studies"

_ijms, 2021, doi:10.3390/ijms22115541_

Round 1
Reviewer 1 Report
The manuscript is the further improved version of the previous one reviewed earlier. The manuscript provides interesting data that might be the basis of further investigations.
The overall merit makes the manuscript suitable for publication.
However, some minor points might be improved by authors.
In the introduction the effects of statins on cancer cells is mentioned (lines 59-61); however, it should be considered that such effects (as well as other responses reported in other papers) were observed in cultured cells, but not -to my knowledge- in vivo, where the cholesterol availability and supply from the bloodstream and surrounding tissues should be considered, which would make tumour cells actually independent from the cholesterol synthesis. I'd encourage authors to include such concepts.
Moreover, if the authors think that their data could be helpful for the design of more specific AK inhibitor, this might be discussed. Similarly, a more comprehensive discussion of the cases where the inhibition of AK might result in clinically significant effect would improve the manuscript.
Author Response
Response to Review Comments
(Manuscript number: ijms-1226165)
The authors are thankful to the Reviewer for valuable for review. We have revised our present manuscript according to Reviewer suggestions and comments. All corrections were applied in the manuscript using the "Track Changes" function. Our point-by-point responses are given below.
Response to Reviewer 1 Comments
Point 1: In the introduction the effects of statins on cancer cells is mentioned (lines 59-61); however, it should be considered that such effects (as well as other responses reported in other papers) were observed in cultured cells, but not-to my knowledge- in vivo, where the cholesterol availability and supply from the bloodstream and surrounding tissues should be considered, which would make tumour cells actually independent from the cholesterol synthesis. I'd encourage authors to include such concepts.
Response 1: In the Introduction section, we have provided both in vivo and in vitro findings for statin pleiotropy, however the in vivo findings might be not highlighted enough. We corrected it in the Introduction section.
We agree with the Reviewer’s concept that the cholesterol availability and supply from bloodstream and surrounding tissue may have an impact on observed statins effects. However, here we would like to point out that many of these so-called pleiotropic effects are cholesterol-independent, and have been shown to be secondary to the inhibition of the synthesis of isoprenoid intermediates of the mevalonate pathway. We think that the present version of the Introduction provides enough information about statins, their pleiotropy and potential therapeutic use to provide enough scientific background for the Readers in the context of our studies and also to discuss a possible link of discovered statin-hAK1 interactions with the pleiotropic and side effects of these drugs.
Point 2: Moreover, if the authors think that their data could be helpful for the design of more specific AK inhibitor, this might be discussed.
Response 2: Thank you for this suggestion. In the Results and Discussion section (Lines: 346-351), we discussed design of specific inhibitors.
Point 3: Similarly, a more comprehensive discussion of the cases where the inhibition of AK might result in clinically significant effect would improve the manuscript.
Response 3: Thank you for this suggestion. In the Results and Discussion section (Lines: 358-380), we discussed in more detail the possible medical implications of hAK1 inhibition by statins.

Reviewer 2 Report
The manuscript is newer up brings a new repurpose of statins as inhibitors of short type adenylate kinases. Adds significant information in the respective fields. Few moderate figures and tables need to be added wherever required. English needs to be improved. The manuscript can be accepted after few minor improvements.
Author Response
Response to Review Comments
(Manuscript number: ijms-1226165)
The authors are thankful to the Reviewer for valuable for review. We have revised our present manuscript according to Reviewer suggestions and comments. All corrections were applied in the manuscript using the "Track Changes" function. Our point-by-point responses are given below.
Response to Reviewer 2 Comments
Point 1: Few moderate figures and tables need to be added wherever required.
Response 1: We added emission spectra of hAK1 in the presence of different concentrations of statin (at 20 and 25°C) in the Figure S2, and provided fluorescence spectra for control samples in the Figure S3.
Point 2: English needs to be improved.
Response 2: We carefully went through the whole manuscript text and introduced corrections to improve English language and style.

Reviewer 3 Report
The authors present a nice and valuable study on statin interaction with hAK1, however, I find one strong weak point limiting the overall evaluation of the study. I am wondering, why have the authors chosen only 5 out of 8 available statins, why did they not use also cerivastatin, which is a very potent statin as well as e.g. pitavastatin? The study would be much more valuable when a comparison of all statins would be made. I suggest adding the missing statins, otherwise, it is fully unclear based on what did the authors selected the five statins used in the study.
At the beginning of the results, some rationale based on which the enzyme for this study has been selected is missing
In Figure 3, controls for the experiment are missing, the authors must use controls.
The authors should explain, why 20-25°C was used for the measurement together with the understandable physiologically relevant temperature of 37°C. Please add some information on the temperatures used.
in the introduction, at lines 47 and 60, two very important references covering statin anticancer activity, are missing:
Variability in statin-induced changes in gene expression profiles of pancreatic cancer. Sci Rep. 2017 Mar 9;7:44219. doi: 10.1038/srep44219.
Isoprenoids responsible for protein prenylation modulate the biological effects of statins on pancreatic cancer cells. Lipids in Health and Disease Volume 16, Article number: 250 (2017)
Author Response
Response to Review Comments
(Manuscript number: ijms-1226165)
The authors are thankful to the Reviewer for valuable for review. We have revised our present manuscript according to Reviewer suggestions and comments. All corrections were applied in the manuscript using the "Track Changes" function. Our point-by-point responses are given below.
Response to Reviewer 3 Comments
Point 1: The authors present a nice and valuable study on statin interaction with hAK1, however, I find one strong weak point limiting the overall evaluation of the study. I am wondering, why have the authors chosen only 5 out of 8 available statins, why did they not use also cerivastatin, which is a very potent statin as well as e.g. pitavastatin? The study would be much more valuable when a comparison of all statins would be made. I suggest adding the missing statins, otherwise, it is fully unclear based on what did the authors selected the five statins used in the study.
Response 1: Thank you for this comment. In the Introduction, we wrote that there are two types of statins (Line 81-83), however we admit that we did not clarify our study concept and statins choice. In our study, we chose representatives of both types of statins based on some clear structural differences, e.g. for type 1, lactone and hydroxy acid form. We did not decide to use cerivastatin since it was withdrawn from the market in 2001 (Furberg and Pitt, 2001, doi: 10.1186/cvm-2-5-205). Whereas all type 2 statins are as hydroxy acid form, and we chose in total 3 representatives. It was shown that, on the basis of data related to chemical structure and physicochemical properties, pitavastatin and fluvastatin are highly similar, while atorvastatin and rosuvastatin are the most different from other statins, therefore they were interesting candidates (Dreyer et al., 2018, doi.org/10.1515/bams-2018-0034;). In the Results and Discussion section 2.2, we explained our choice of statins (Line: 163-170).
Point 2: At the beginning of the results, some rationale based on which the enzyme for this study has been selected is missing.
Response 2: At the beginning of the results, we added a short rationale as required (Lines: 139-145). We would like to kindly point out that the detailed rationale why we selected this AK enzyme is provided in the Introduction: Lines: 95-99 and 110-134.
Point 3: In Figure 3, controls for the experiment are missing, the authors must use controls.
Response 3: Thank you for this remark. In the Supplemental Figure S3, we provided the required controls.
Point 4: The authors should explain, why 20-25°C was used for the measurement together with the understandable physiologically relevant temperature of 37°C. Please add some information on the temperatures used.
Response 4: Thank you for this remark. In our study we determined the binding ability of the statins to human AK1 at three temperature. We used temperature of 37°C, which, as Reviewer mentioned, is the physiological temperature. Moreover, we wanted to evaluate the effect of temperature changes on the binding constant. We chose temperature 25°C and we observed that the Kb values are not markedly different from values determined at 37°C. Therefore, as the third temperature, we chose a lower temperature (20°C). The Kb values for all statins at 20°C turned out the lowest values and markedly different from the others measured at 25 and 37°C. The obtained results allowed us to conclude that the stability of the complex increases with the temperature decrease. We corrected the manuscript as requested.
Point 5: in the introduction, at lines 47 and 60, two very important references covering statin anticancer activity, are missing:
Variability in statin-induced changes in gene expression profiles of pancreatic cancer. Sci Rep. 2017 Mar 9;7:44219. doi: 10.1038/srep44219.
Isoprenoids responsible for protein prenylation modulate the biological effects of statins on pancreatic cancer cells. Lipids in Health and Disease Volume 16, Article number: 250 (2017)
Response 5: We are thankful for this valuable remark. We have described the most important findings from these studies and added required references (Introduction section).

Round 2
Reviewer 3 Report
The authors have answered what was asked and improved their article correspondingly, therefore, now I recommend the article for publication.
This manuscript is a resubmission of an earlier submission. The following is a list of the peer review reports and author responses from that submission.
Round 1
Reviewer 1 Report
The research submitted is on assessing the interactions between human adenylate kinase isoenzyme 1 (hAK1) and atorvastatin (AVS), fluvastatin (FVS), pravastatin (PVS), rosuvastatin (RVS) and simvastatin (SVS) with fluorescence spectroscopy, and determined the impact of these drugs on the hAK1 enzymatic activity by HPLC that is newer and limited number of reports have been published till date. The research has topics bringing up the importance and need for understanding the molecular aspects and their influence right from synthesis, delivery to ADME in the body & also it rationalises majority of interactions, pathways that have been reported to date with statins. The figures and tables depicted are in sync with the content. But text embedded in all the figures need to be improved for readability. The research is well articulated but further needs more information and modifications as per the following recommendations. The authors are even advised for English language revisions as well. These below major revisions are suggested.
- Title seems to be very in direct, It can be “Assessing the interaction of statin with human AK1….. Authors are strongly suggested for this change.
- Abstract is too short and indirect. Authors need to elevate the importance of understanding the importance of statin interactions – How these pathways can be promising than available approaches for drug targeting. Authors are suggested to mention the objective and importance of this research in abstract.
- Keywords are not strong enough. Few repetitive needs to be replaced. Single words are encouraged.
- Introduction has too big sentences need to be short and concise.
- Statistical significance p value is not denoted in Figure 5.
- Conclusion seems half informed, authors need to elevate the outlook of the statins and how understanding the interactions with enzymes are able to address and limit its adverse effects. Changes are advised.
- Repurposing of drugs is a futuristic approach to address the unmet clinical needs. Apart from anti-hyperlipidemic activity Statins have been explored for anti-cancer, bone growth and for treatment of various other ailments. This section is completely ignored in the introduction section submitted. Authors are strongly recommended to citing useful and latest references.
- The present review touches upon the clinical performance and limitations of all forms from statin family. However, the novel delivery approaches that have been reported for improvement of physiochemical properties, dissolution that have direct effect on bioavailability and in-vivo performance have been neglected. The authors are strongly recommended to cite only the latest references not in depth.
- Points addressed in 7 & 8 should be touched in the abstract as well.
- In support of points raised in 7 & 8, Authors are strongly advised to mention and cite the following repurposing of statins and latest novel delivery systems prepared and reported in literature. Citing all these below references in introduction and in other relevant separate sections is highly recommended. This could elevate the importance of the review to the readers and support the benefit of this review if accepted or published online.
Tumor Biology 42, no. 7 (2020): 1010428320941760
International Journal of Pharmaceutics, p.119534.
Journal of Pharmaceutical Investigation (2020): 1-17.
Journal of Drug Delivery and Therapeutics, 3(3), pp.131-140.
Biochimica et Biophysica Acta (BBA)-Biomembranes (2020): 183306.
International journal of nanomedicine 10 (2015): 321.
Wound Repair and Regeneration (2020).
Current Drug Delivery 13, no. 2 (2016): 211-220.
Nanoscale, 12(17), pp.9541-9556.
Marine drugs 18, no. 4 (2020): 226.
Journal of Applied Pharmaceutical Science, 10(09), pp.001-011.
International Journal of Pharmaceutics (2020): 119438.
Journal of Pharmacy and Pharmacology, 69(6), pp.613-624.
Marine Drugs 18, no. 4 (2020): 201.
Drug discovery today, 24(2), pp.567-574.
Author Response
Response to Review Comments
(Manuscript number: ijms-1019370)
The authors are very thankful to the Reviewer for their deep and thorough review. We have revised our manuscript according to the Reviewer’ suggestions and comments. We hope our revision has improved the manuscript to a level of their satisfaction. We have addressed point-by-point all issues raised by Reviewer. All corrections were implemented as suggested and highlighted in yellow in the revised manuscript.
Response to Reviewer 1 Comments
Point 1. The figures and tables depicted are in sync with the content. But text embedded in all the figures need to be improved for readability.
Response 1: After careful revision of the figures, we improved the text for better readability in Figures 1, 3, 5, 6.
Point 2: Title seems to be very in direct, It can be “Assessing the interaction of statin with human AK1….. Authors are strongly suggested for this change.
Response 2: We changed the title according to the Reviewer suggestion. The new title is as follows: “Assessing the interactions of statins with human adenylate kinase isoenzyme 1: fluorescence and enzyme kinetic studies”
Point 3: Abstract is too short and indirect. Authors need to elevate the importance of understanding the importance of statin interactions – How these pathways can be promising than available approaches for drug targeting. Authors are suggested to mention the objective and importance of this research in abstract.
Response 3: We improved the abstract by highlighting the importance of studying statin-enzyme interactions. The abstract was also revised to clearly emphasize the objective and importance of our research.
Point 4: Keywords are not strong enough. Few repetitive needs to be replaced. Single words are encouraged.
Response 4: Due to Reviewer suggestion, keywords have been corrected as follow:
adenylate kinase, atorvastatin, fluvastatin, pravastatin, rosuvastatin, simvastatin, inhibitors, fluorescence spectroscopy, pleiotropic effect
Point 5. Introduction has too big sentences need to be short and concise.
Response 5: Introduction was carefully revised for the language and sentence structure.
Point 6: Statistical significance p value is not denoted in Figure 5.
Response 6: According to the remark, the p values have been denoted in Figure 5.
Point 7: Conclusion seems half informed, authors need to elevate the outlook of the statins and how understanding the interactions with enzymes are able to address and limit its adverse effects. Changes are advised.
Response 7: As suggested, in the Conclusions section, we have discussed possible medical implications of discovered statin-hAK1 interactions (lines: 436-446).
Point 8: Repurposing of drugs is a futuristic approach to address the unmet clinical needs. Apart from anti-hyperlipidemic activity Statins have been explored for anti-cancer, bone growth and for treatment of various other ailments. This section is completely ignored in the introduction section submitted. Authors are strongly recommended to citing useful and latest references.
Response 8: Thank you for your valuable suggestion. In the introduction section, the pleiotropic effect of statins are now described in more details (lines: 44-56).
Point 9: The present review touches upon the clinical performance and limitations of all forms from statin family. However, the novel delivery approaches that have been reported for improvement of physiochemical properties, dissolution that have direct effect on bioavailability and in-vivo performance have been neglected. The authors are strongly recommended to cite only the latest references not in depth.
Response 9: The Reviewer's suggestion was applied in the revised version of the manuscript. We discussed novel delivery approaches for the improvement of the bioavailability and in vivo efficacy of statins (lines: 56-61).
Point 10: Points addressed in 7 & 8 should be touched in the abstract as well.
Response 10: As recommended, we provided the information on the statin pleiotropy and novel drug delivery approaches.
Point 11: In support of points raised in 7 & 8, Authors are strongly advised to mention and cite the following repurposing of statins and latest novel delivery systems prepared and reported in literature. Citing all these below references in introduction and in other relevant separate sections is highly recommended. This could elevate the importance of the review to the readers and support the benefit of this review if accepted or published online.
Response 11: We are thankful to the Reviewer for providing us the references. We included the majority of them.

Reviewer 2 Report
Authors report experiments that highlight the inhibitory effect of statins on hAK.
The manuscript is well written and provides new data about the pleiotropic effects of this widely used drug class.
However, I would encourage authors to improve the introduction and the discussion and/or conclusions. In particular, the introduction should include a more extensive description of AK and its biological role and importance. Similarly, information and consideration about the practical perspectives and biological/medical implications of the data reported should be included in the discussion and/or conclusions.
Furthermore, authors should consider the possibility of extending the introduction addressing more explicitly and comprehensively the topic of pleiotropic effects (shortly reported in lines 43-50). Even if not directly related to the focus of the manuscript, providing this information would improve the quality of the introduction.
Moreover, as an added value, it would be very interesting to evaluate and discuss if any similarity in protein-drug interaction does exist with other enzymes involved in other described pleiotropic effects.
Author Response
Response to Review Comments
(Manuscript number: ijms-1019370)
The authors are very thankful to the Reviewer for their deep and thorough review. We have revised our manuscript according to the Reviewer’ suggestions and comments. We hope our revision has improved the manuscript to a level of their satisfaction. We have addressed point-by-point all issues raised by Reviewer. All corrections were implemented as suggested and highlighted in yellow in the revised manuscript.
Response to Reviewer 2 Comments
Point 1: In particular, the introduction should include a more extensive description of AK and its biological role and importance. Similarly, information and consideration about the practical perspectives and biological/medical implications of the data reported should be included in the discussion and/or conclusions.
Response 1: We thank the Reviewers for raising this point. We discussed a biological role and importance of hAK1 in the Introduction/Conclusions sections and added appropriate references (lines: 90-103 and 436-446).
Point 2: Furthermore, authors should consider the possibility of extending the introduction addressing more explicitly and comprehensively the topic of pleiotropic effects (shortly reported in lines 43-50). Even if not directly related to the focus of the manuscript, providing this information would improve the quality of the introduction.
Response 2: Thank you for your suggestion. In the introduction, we described the pleiotropic effect of statins in detail (lines: 44-56).
Point 3: Moreover, as an added value, it would be very interesting to evaluate and discuss if any similarity in protein-drug interaction does exist with other enzymes involved in other described pleiotropic effects.
Response 3: To the best of our knowledge, the reported statin-protein interactions include enzymes of the cytochrome P450 family (involved in drug metabolism and bio-activation), and two proteins that we mentioned in the Introduction section: β2 integrin leukocyte function antigen-1 (LFA-1) and P-glycoprotein (P-gp). We agree that studies on possible interactions between statins and other enzymes, e.g. other enzymes metabolizing nucleotides would be very interesting. This will be within the scope of our future studies.

Round 2
Reviewer 2 Report
The authors improved the manuscript by addressing the reviewers' requests, although more details could have been added in the introduction, especially about the anticancer effects of statins. Conversely, in my opinion, the information in lines 56-65, dealing with pharmaceutic technologies of statins formulation could be omitted (they are poorly related to the topic of the manuscript).
Note: figure 1 is not visualized (a black box appears), at least on my computer.
Author Response
Response to Review Comments
(Manuscript number: ijms-1019370)
The authors are very thankful to the Reviewer for further comments. We have revised our manuscript and carefully addressed to the Reviewer’s remarks. All new corrections were implemented as suggested and highlighted in green in the revised manuscript.
Response to Reviewer 2 Comments
Point 1: The authors improved the manuscript by addressing the reviewers' requests, although more details could have been added in the introduction, especially about the anticancer effects of statins.
Response 1: In the introduction section, we described in more details the mechanisms underlying pleiotropic effects of statins, including the anticancer effects.
Point 2: Conversely, in my opinion, the information in lines 56-65, dealing with pharmaceutic technologies of statins formulation could be omitted (they are poorly related to the topic of the manuscript).
Response 2: As strongly recommended by the Reviewer 1, we decided to add a brief information on the recent advances in improving the pharmacological properties of statins. We agree that this topic is not firmly related to the manuscript, however it is relevant in the context of repurposing these drugs to treat other diseases what we described in the Introduction. This research field is developing and relevant in view of the evolving concept for the management of diverse diseases with statins.
Point 3: Note: figure 1 is not visualized (a black box appears), at least on my computer.
Response 3: The Figure 1 was corrected.
